# Kounis Syndrome in Clinical Practice: Insights from Clinical Case Series and Mechanistic Pathways

**DOI:** 10.3390/jcm14030768

**Published:** 2025-01-24

**Authors:** Laura-Cătălina Benchea, Larisa Anghel, Dragoș Viorel Scripcariu, Anca Diaconu, Răzvan-Liviu Zanfirescu, Laurentiu-Vladimir Lucaci, Silviu-Gabriel Bîrgoan, Radu Andy Sascău, Cristian Stătescu, Rodica Radu

**Affiliations:** 1Internal Medicine Department, “Grigore T. Popa” University of Medicine and Pharmacy, 700503 Iași, Romania; benchea.laura-catalina@d.umfiasi.ro (L.-C.B.); vladimir.lucaci@umfiasi.ro (L.-V.L.); radu.sascau@umfiasi.ro (R.A.S.); cristian.statescu@umfiasi.ro (C.S.); rodica.radu@umfiasi.ro (R.R.); 2Cardiology Department, Cardiovascular Diseases Institute “Prof. Dr. George I. M. Georgescu”, 700503 Iași, Romania; anca.a.diaconu@gmail.com (A.D.); zanfirescu_razvan-liviu@d.umfiasi.ro (R.-L.Z.); birgoan_gabi@yahoo.com (S.-G.B.); 3Department of Surgical Sciences, University of Medicine and Pharmacy “Grigore T. Popa”, 700115 Iasi, Romania; dscripcariu@gmail.com; 4Pathophysiology Department, ”Grigore T. Popa” University of Medicine and Pharmacy, 700503 Iași, Romania

**Keywords:** Kounis syndrome, hypersensitivity reactions, acute coronary syndrome, coronary vasospasm, multidisciplinary management

## Abstract

Kounis syndrome (KS) is a rare condition where hypersensitivity reactions trigger coronary vasospasm, destabilization of atherosclerotic plaques, or stent thrombosis, posing diagnostic and therapeutic challenges due to its overlap with acute coronary syndrome (ACS) and the absence of specific guidelines. This study reviews cases of KS from the Institute of Cardiovascular Disease to highlight clinical presentations, triggers, and treatment strategies. We analyzed four cases of KS treated at our institution between 2019 and 2024. Detailed clinical histories, laboratory findings, imaging studies, and treatment plans were reviewed. Patients were classified by KS subtype based on coronary anatomy and pathophysiological mechanisms. Management strategies were tailored to each subtype, combining myocardial revascularization, antiplatelet therapy, and treatment for allergic reactions. The series included two cases of Type I KS in patients with structurally normal coronary arteries and two cases of Type II KS involving pre-existing atherosclerosis. No Type III KS was observed. Triggers included insect stings, antibiotics, iodinated contrast agents, and anesthetics. Coronary angiography confirmed the diagnosis in all cases. Treatments included percutaneous coronary interventions, dual antiplatelet therapy, and prophylactic antihistamines or corticosteroids. All patients experienced favorable outcomes, although diagnostic delays were noted in cases with atypical presentations. KS remains underdiagnosed, especially in emergency settings where it mimics ACS. Early recognition and multidisciplinary management involving allergology and cardiology are crucial. Future research should focus on safer diagnostic tools, understanding the pathophysiology, and developing evidence-based preventive strategies. Increasing the awareness of KS and its inclusion in ACS differentials are essential to improving patient outcomes and preventing recurrences.

## 1. Introduction

Kounis syndrome (KS) is a complex clinical entity at the crossroads of allergic responses and acute coronary syndrome (ACS), where hypersensitivity reactions trigger coronary vasospasm, atherosclerotic plaque destabilization, or thrombosis. Although the syndrome’s presentation often overlaps with typical ACS, the coexistence of allergic symptoms, such as urticaria, angioedema, or respiratory distress, suggests an allergic etiology [1]. Despite its increasing recognition, KS remains underdiagnosed or misdiagnosed, largely due to the lack of awareness and specific guidelines for its management. While the exact incidence of KS is not well defined due to underreporting, studies suggest that it may affect up to 1–2% of patients who present with acute coronary syndrome accompanied by allergic symptoms. There are currently limited studies in the literature on this topic, as it is a rare condition. The rarity of KS has limited large-scale studies, with most insights derived from isolated case reports or small case series. Understanding the pathophysiology, classification, and clinical presentations of KS is vital for timely diagnosis and treatment, particularly in emergency settings where the condition is most likely to manifest [2].

## 2. Classification and Pathophysiological Mechanism

Kounis syndrome represents a multifaceted interaction between allergic responses and acute coronary syndrome, wherein hypersensitivity reactions may lead to coronary vasospasm, destabilization of atherosclerotic plaques, or thrombosis within coronary stents [2].

Current classification outlines KS into three subtypes, with recent literature providing refined insights into the mechanisms underlying each type: Type I KS: allergic vasospastic angina in patients without coronary artery disease (CAD); Type II KS: acute myocardial infarction in patients with pre-existing atherosclerosis; and Type III KS: stent thrombosis from hypersensitivity reactions to stent materials (Table 1) [1]. Kounis syndrome is a condition where cardiovascular and immunological processes converge, driven by mast cell degranulation, which releases vasoactive mediators like histamine and platelet-activating factor, leading to coronary vasospasm, plaque destabilization, or thrombosis. Its classification into subtypes reflects distinct pathophysiological mechanisms influenced by the patient’s coronary anatomy and immune response [3].

Type I KS is observed in individuals with structurally normal coronary arteries and is characterized by coronary vasospasm without the involvement of atherosclerotic plaques. Upon allergen exposure, mast cells undergo degranulation, releasing mediators such as histamine, leukotrienes, and platelet-activating factor (PAF). These substances act on vascular smooth muscle cells, leading to coronary spasm. Research indicates that histamine receptors on coronary smooth muscle play a significant role in this process, identifying histamine as a key mediator in Type I KS. This phenomenon often occurs in younger individuals or those without conventional cardiovascular risk factors. Recent research underscores the influence of mechanical forces on mast cell activation, highlighting the interaction between cellular responses and the extracellular matrix. Variations in vascular matrix stiffness and activation of mechanosensitive signaling pathways, such as the mitogen-activated protein kinase (MAPK) and integrin-mediated pathways, may modulate the degree of mast cell degranulation and its subsequent cardiovascular effects. These advancements provide deeper insight into the intricate mechanisms of hypersensitivity reactions and their role in the development of Kounis syndrome [4]. Emerging evidence has highlighted the capacity of mast cells to undergo activation and degranulation through non-IgE-mediated mechanisms, commonly referred to as pseudoallergic reactions. These pathways involve direct stimulation of mast cell receptors by various agents, including radiocontrast agents, drugs, and specific peptides. Unlike classical IgE-mediated Type 1 hypersensitivity reactions, pseudoallergies bypass the need for IgE antibodies yet still result in the release of potent inflammatory mediators such as histamine, leukotrienes, and platelet-activating factor. This non-IgE-mediated mast cell activation is increasingly recognized as a potential contributor to the pathogenesis of Kounis syndrome, particularly in scenarios where traditional IgE involvement is absent, underscoring its clinical relevance in understanding and managing hypersensitivity-related coronary events. This finding expands the range of allergens capable of precipitating Type I KS and underscores its complex pathophysiology [3,4]. Type II KS presents as an acute myocardial infarction triggered by allergen-induced destabilization of atherosclerotic plaques. Immune mediators released during an allergic reaction contribute to plaque instability, culminating in rupture and subsequent thrombotic complications. Recent studies highlight the role of enzymes such as tryptase and chymase, which are released during mast cell degranulation. These enzymes degrade structural proteins like collagen within the extracellular matrix of the plaque, weakening the fibrous cap and transforming a stable plaque into a vulnerable one. Additionally, platelet-activating factors and other pro-thrombotic mediators enhance platelet aggregation and promote thrombus formation at the site of plaque rupture, worsening ischemic outcomes. This inflammatory cascade increases the likelihood of acute coronary syndrome in patients with underlying atherosclerosis. Research, including the findings of Ollo-Morales et al., emphasizes the role of mast cell products in exacerbating plaque vulnerability, predisposing it to rupture [2]. These findings underscore the importance of recognizing allergic reactions as potential triggers for acute coronary events in patients with underlying coronary artery disease [4]. Type III KS in patients with coronary stents arises from an allergic reaction to stent materials or drug-eluting coatings, leading to localized inflammation and thrombosis. This variant is particularly concerning due to hypersensitivity to metals such as nickel or polymers in drug-eluting stents, which triggers eosinophilic infiltration and pronounced inflammatory responses at the stent site [5]. Histopathological studies reveal that eosinophils, alongside other immune cells, accumulate around the thrombus, intensifying localized thrombosis and causing significant ischemia. Management often necessitates targeted antiplatelet and anti-inflammatory therapies to address both the allergic and thrombotic aspects of the condition. Recognizing this mechanism is crucial for cardiologists treating patients with known sensitivities to stent components or drug coatings [6].

Thus, Kounis syndrome (KS) is characterized by a complex interplay of immunological and cardiovascular mechanisms involving both IgE-mediated and non-IgE-mediated pathways. In the classic IgE-mediated response, allergens bind to IgE antibodies on the surface of mast cells and basophils, leading to degranulation and the release of vasoactive mediators such as histamine, platelet-activating factor, and leukotrienes. Recent studies have also highlighted non-IgE-mediated mechanisms, where allergens directly activate mast cells without prior sensitization, broadening the spectrum of potential triggers to include both typical allergens like food and insect venom and atypical ones such as environmental toxins [1].

Upon exposure to an allergen, mast cells and other immune cells release inflammatory mediators that induce vasoconstriction, increase vascular permeability, and promote platelet aggregation. These processes lead to coronary artery spasm, plaque destabilization through enzymatic degradation of the extracellular matrix, and thrombosis, particularly in the context of coronary stents. Together, these mechanisms underscore how diverse immunological triggers can precipitate significant cardiac events in susceptible individuals [1]. Thus, Kounis syndrome involves mast cell activation through both IgE-mediated and non-IgE-mediated pathways, leading to the release of inflammatory mediators that trigger cardiovascular events. IgE-mediated reactions occur following sensitization to a specific allergen, where re-exposure causes cross-linking of IgE antibodies on mast cells, inducing degranulation. By contrast, non-IgE-mediated mechanisms, such as pseudoallergies, involve direct activation of mast cells by stimuli like drugs or physical factors, independent of prior sensitization or IgE involvement [1,2].

## 3. Clinical Perspectives

The purpose of this study is to enhance the understanding of Kounis syndrome by combining a presentation of four clinically documented cases with a comprehensive review of the existing literature. These cases, managed at the Institute of Cardiovascular Disease in Eastern Romania between 2019 and 2024, illustrate the diverse triggers, clinical manifestations, and outcomes associated with KS. The cases include two instances of Type I KS in patients without obstructive coronary artery disease and two cases of Type II KS in individuals with preexisting atherosclerosis, with identified triggers such as pharmacological agents and insect bites. By integrating these real-world clinical experiences with a review of the broader literature, the study aims to shed light on the mechanisms, diagnostic challenges, and management strategies of KS, a rare but potentially severe condition that requires prompt recognition and tailored intervention.

### 3.1. Case 1

A 75-year-old male, an ex-smoker with multiple cardiovascular risk factors, including grade I obesity, hypertension, and dyslipidemia, was referred to our clinic following a postero-infero-lateral ST-segment elevation myocardial infarction (STEMI) (Figure 1). Symptoms of angina began five hours before presentation and were preceded by a bee sting, which triggered a syncope. The patient was undergoing immunomodulatory therapy for rheumatoid arthritis, including methotrexate, adalimumab, and tamsulosin, and had a history of atopy and anaphylactic shock secondary to a bee sting. On admission, the patient was hemodynamically and respiratory stable, with a blood pressure of 144/75 mmHg, a heart rate of 77 bpm, and oxygen saturation (SpO_2_) of 99%. Laboratory investigations revealed elevated myocardial enzymes, mild renal impairment (eGFR = 55 mL/min/1.73 m^2^), and systemic inflammatory markers, including leukocytosis (13,620/mm^3^) with eosinophilia (9.5%), elevated C-reactive protein (CRP = 17.5 mg/L), and IgE levels (180 UI/mL). Coronary angiography identified an acute thrombotic occlusion of the posterior interventricular artery, with no evidence of other atherosclerotic lesions. The occlusion was successfully treated with coronary angioplasty and the placement of a drug-eluting stent (Figure 2). The patient was initiated on dual antiplatelet therapy (aspirin 75 mg once daily and ticagrelor 90 mg twice daily), high-dose statin therapy (atorvastatin 80 mg once daily), an ACE inhibitor (ramipril 5 mg once daily), and a beta-blocker (bisoprolol 2.5 mg once daily). The patient’s clinical course was favorable, with discharge occurring seven days after admission. At discharge, he had a mildly reduced ejection fraction but experienced no further complications.

### 3.2. Case 2

A 42-year-old male with cardiovascular risk factors, including smoking, grade I obesity, and dyslipidemia, presented to our clinic with an inferolateral ST-segment elevation myocardial infarction. The patient had no prior history of allergic reactions or significant personal or family medical history. However, detailed anamnesis revealed that symptoms began approximately one hour after the ingestion of a 1 g tablet of amoxicillin/clavulanic acid during a dental consultation. On clinical examination, the patient had a blood pressure of 130/100 mmHg, heart rate of 86 bpm, and oxygen saturation (SpO_2_) of 99%. Laboratory investigations demonstrated elevated myocardial enzymes (CK-MB = 128 U/L, LDH = 1390 U/L, AST = 175 U/L), marked dyslipidemia (non-HDL cholesterol = 201 mg/dL, LDL cholesterol = 145 mg/dL), and evidence of an inflammatory response (CRP = 105 mg/L, erythrocyte sedimentation rate = 46 mm/h, fibrinogen = 660 mg/dL). Echocardiography revealed a normal left ventricular ejection fraction (LVEF = 55%) with hypokinesia of the inferior and lateral walls. Coronary angiography showed an acute thrombotic occlusion of the proximal right coronary artery. The lesion was treated with percutaneous coronary angioplasty and the deployment of two drug-eluting stents (Figure 3). However, no reflow phenomena were observed during the procedure, necessitating the initiation of triple antithrombotic therapy, including aspirin (75 mg once daily), clopidogrel (75 mg once daily), and acenocoumarol with strict International Normalized Ratio (INR) monitoring for a limited duration.

### 3.3. Case 3

A 43-year-old woman was transferred to our hospital from a Surgical Oncology clinic after experiencing a lateral ST-segment elevation myocardial infarction during anesthesia induction for the treatment of endometrial adenocarcinoma. During induction, the patient developed angioedema, which was promptly treated by the oncology team. She had no known history of allergies and was undergoing hormonal replacement therapy for hypothyroidism. On admission, her hemodynamic and respiratory parameters were stable, with a blood pressure of 100/70 mmHg, heart rate of 78 bpm, and oxygen saturation (SpO_2_) of 98%. Laboratory findings showed significantly elevated high-sensitivity troponin I levels (6.3 mg/L), leukocytosis (12,400/mm^3^) with eosinophilia (10.2%), and mildly elevated C-reactive protein (6.4 mg/L), indicative of systemic inflammation. Her lipid profile revealed dyslipidemia with elevated LDL cholesterol (159 mg/dL) and non-HDL cholesterol (171 mg/dL). Echocardiography demonstrated normal left ventricular systolic function (LVEF = 55%) but revealed hypokinesia of the lateral wall. Coronary angiography showed normal coronary arteries, leading to the diagnosis of myocardial infarction with non-obstructive coronary arteries (MINOCA) (Figure 4). The likely etiology was attributed to coronary vasospasm induced by atracurium administration during anesthesia induction. The patient was discharged with a prescription for a low-dose calcium channel blocker to prevent recurrence.

### 3.4. Case 4

A 68-year-old male with a history of cardiovascular risk factors, including hypertension, diabetes, dyslipidemia, and prior balloon angioplasty of the left popliteal artery in 2012, was admitted to our clinic for elective angiography of peripheral arteries due to recurrent intermittent claudication. Given a recent diagnosis of heart failure with reduced ejection fraction (LVEF = 30%) and left bundle branch block, coronary angiography was also planned. The patient was clinically stable at presentation, with euvolemia and hemodynamic parameters within normal limits (BP = 150/90 mmHg, HR = 90 bpm, SpO_2_ = 95%). Due to a known history of allergic reaction to iodinated contrast agents, which previously manifested as pruritus and an asthma attack, prophylactic intravenous hydrocortisone hemisuccinate was administered before the procedure. Despite this precaution, the patient experienced asystole immediately following the administration of iodinated contrast agent into the right coronary artery. Resuscitation with chest compressions, adrenaline, and artificial ventilation was successful. Coronary angiography revealed no obstructive coronary artery disease, and the cardiac arrest was attributed to coronary artery spasm secondary to an allergic reaction to the contrast agent. The patient was closely monitored in the intensive care unit and showed a favorable recovery, with successful extubation and subsequent discharge after an 8-day hospitalization.

## 4. Allergic and Cardiac Interplay

KS presents with a range of symptoms similar to traditional acute coronary syndrome, such as chest pain and shortness of breath, often accompanied by allergic signs like urticaria or angioedema. Recognizing these overlapping symptoms with ACS is critical, particularly in patients who have recently encountered known allergens [7].

KS presents with a combination of allergic and cardiac symptoms, making diagnosis challenging. Common clinical manifestations include chest pain, dyspnea, urticaria, and angioedema. Electrocardiographic changes may mimic those of acute coronary syndrome, and elevated cardiac biomarkers can be observed. A thorough patient history, including recent exposure to potential allergens, is essential. Elevated serum tryptase levels can support the diagnosis by indicating mast cell activation [8].

Kounis syndrome is frequently accompanied by other systemic manifestations of an allergic reaction, reflecting its basis in hypersensitivity mechanisms. The literature suggests that up to 70% of patients with KS present with extracardiac signs of allergy, such as skin rash, urticaria, flushing, or angioedema. These manifestations are often the initial indicators of an allergic event and can provide crucial diagnostic clues in the context of KS. These symptoms are frequently triggered by exposure to specific allergens, such as medications, foods, or environmental agents, adding complexity to the syndrome’s clinical profile. The presence of such symptoms highlights the systemic nature of the hypersensitivity response and reinforces the need for a comprehensive evaluation of allergic triggers in patients presenting with cardiac symptoms indicative of KS [2].

Type I Kounis syndrome predominantly occurs in patients with no pre-existing coronary artery disease. It is characterized by transient episodes of chest pain caused by coronary vasospasm, which is triggered by the release of allergic mediators such as histamine and leukotrienes. This episodic chest pain reflects the reversible nature of the vasospasm in structurally normal coronary arteries. Accurate differentiation from other non-ischemic causes of chest pain is crucial for proper diagnosis and management [3].

Type II Kounis syndrome closely mimics an acute myocardial infarction (AMI) and occurs in individuals with pre-existing coronary artery disease. In this subtype, an allergic response exacerbates atherosclerotic plaque instability, leading to plaque rupture. This process results in prolonged and severe chest pain, accompanied by significant ECG changes and elevated cardiac biomarkers, indicative of myocardial injury. Due to its clinical similarity with traditional myocardial infarction, Type II KS can be misdiagnosed unless a detailed history of recent allergen exposure or allergic reactions is obtained [2].

Type III Kounis syndrome is characterized by in-stent thrombosis resulting from a hypersensitivity reaction to materials used in coronary stents, such as nickel or drug-eluting polymers. This hypersensitivity triggers a localized inflammatory response, leading to thrombus formation, which significantly obstructs blood flow and is often accompanied by systemic allergic symptoms like rash or angioedema. By contrast, common stent thrombosis is typically due to mechanical or pharmacological factors, such as stent malposition or therapy interruption, without allergic involvement. Not all stent thrombosis cases are related to Type III KS, making accurate differentiation crucial for appropriate management. The resulting ischemic symptoms are typically sudden and severe, requiring immediate intervention to restore coronary perfusion. Prompt recognition of this rapid progression to ischemia, particularly when accompanied by allergic manifestations, is critical for effective and potentially life-saving management [3].

Recognizing and correlating allergic symptoms, such as skin rash, respiratory distress, or gastrointestinal upset, with signs of coronary ischemia is essential for accurately diagnosing Kounis syndrome. The connection between allergic reactions and ischemic events highlights the critical need for early recognition and timely intervention. Given that KS can closely mimic more common forms of acute coronary syndrome, it is crucial to adopt a tailored approach that addresses both the allergic and cardiac components of the condition to ensure effective management [1,9].

Differential diagnosis of Kounis syndrome is essential, as its clinical presentation overlaps with conditions such as Takotsubo cardiomyopathy and isolated anaphylaxis. Takotsubo syndrome (TTS) and Kounis syndrome exhibit overlapping symptoms, such as chest pain and transient ischemic changes, but differ significantly in their underlying mechanisms and imaging characteristics (Table 2). TTS is defined by apical or mid-ventricular ballooning with hypercontractile basal segments observed on echocardiography, myocardial strain abnormalities extending beyond a single coronary artery distribution, and markedly reduced global longitudinal strain (GLS). By contrast, KS lacks apical ballooning, with strain abnormalities confined to the territory of the affected coronary artery [3].

Speckle-tracking echocardiography in TTS demonstrates global impairments in longitudinal, circumferential, and radial left ventricular function. KS, however, presents echocardiographic findings that depend on the extent of ischemia, without the distinctive patterns seen in TTS. Although chest pain is a shared clinical feature, the underlying mechanisms differ: TTS arises from catecholamine-induced myocardial stunning, whereas KS is triggered by allergic reactions leading to coronary vasospasm, plaque destabilization, or stent thrombosis [8].

Notably, Type I KS closely resembles Prinzmetal’s angina due to the presence of coronary vasospasm in the absence of atherosclerosis. These differences highlight the importance of thorough differential diagnosis through clinical evaluation, imaging studies, and laboratory findings.

Biomarkers and advanced imaging techniques play a crucial role in diagnosing Kounis syndrome, offering valuable differentiation from other acute coronary syndromes.

Biomarkers: 

Elevated serum tryptase levels, indicative of mast cell degranulation, are highly useful for confirming KS, especially when allergic symptoms coincide with cardiac events. Eosinophilia provides additional support for an allergic etiology. Cardiac biomarkers, such as troponins, are elevated in Type II and Type III KS, reflecting myocardial injury [3].

Imaging Techniques:

Coronary Angiography: Essential for real-time assessment of coronary artery spasm or thrombus, angiography is particularly valuable in diagnosing Type I and Type III KS.

Intravascular Ultrasound (IVUS) and Optical Coherence Tomography (OCT): In cases of suspected plaque instability, these modalities offer detailed visualization of plaque morphology and stability, crucial for Type II KS [1].

Cardiac MRI with Gadolinium Enhancement: This imaging modality can identify myocardial inflammation even in the absence of visible occlusions on angiography, providing critical insights into atypical presentations of KS.

By integrating biomarker analysis and advanced imaging, clinicians can achieve a more precise diagnosis of KS, enabling targeted and effective management [1].

For patients with recurring episodes of Kounis syndrome or a history suggestive of allergen sensitivity, allergy testing is a valuable diagnostic and preventive tool. Techniques such as total and specific IgE assays, skin prick testing, and allergen-specific IgE measurements can help to identify the triggers responsible for hypersensitivity reactions. These findings can guide tailored preventive strategies. However, caution is necessary during testing, as these procedures may provoke allergic reactions in sensitized individuals. To mitigate risks, tests should be conducted under controlled conditions with appropriate medical supervision. For high-risk patients, especially those prone to severe allergic responses, prophylactic use of mast cell stabilizers (e.g., cromolyn sodium) or antihistamines may help to reduce the likelihood of recurrence. Such preventive measures are particularly beneficial when complete avoidance of known allergens is impractical [2].

## 5. Implications for Clinical Practice

Although it appears to be a rare disease, over 200 cases of KS have been described in the literature, some of them manifesting without evident allergic symptoms, suggesting that KS might be either underdiagnosed in clinical practice or underreported. It is of paramount importance to raise the awareness of KS syndrome among cardiologists, especially those working in the emergency department, in order to allow prompt recognition and management of the disease. Known history of allergic reactions and the presence of other allergic manifestations accompanying the acute coronary syndrome (rash, urticaria, angioedema, dyspnea) should raise suspicion of KS. Thorough anamnesis is essential in identifying recent allergic exposure in individuals presenting with cardiac symptoms, especially in those cases when it is not easily apparent/identifiable. Antibiotics and AINS seem to be the most frequently causes, although other drugs, food, and different environmental exposures were also encountered [2]. Our cases were due to exposure to an antibiotic, an anesthetic, an iodinated contrast agent, and an insect sting.

KS typically occurs more frequently in males, which was also the case in our patients (male-to-female ratio of 3:1), although no significant sex-based differences in mortality has been remarked [10]. While the syndrome predominantly affects patients aged between 40 and 70 years old, cases have been reported in all age groups, including a 2-year-old child [11]. Involvement of other vascular territories, such as cerebral and mesenteric arteries, has also been reported [12,13]. Therefore, a thorough clinical evaluation is essential to identify possible concomitant affected vascular territories. A high index of clinical suspicion of KS syndrome should prompt measurement of serum tryptase and IgE antibodies, along with myocardial enzymes such as troponin (preferably high-sensitivity I troponin) and creatine-kinase [14]. The primary differential diagnosis of allergic myocardial infarction is the stress-induced Takotsubo cardiomyopathy. Additionally, younger patients should be investigated for other causes of myocardial injury and inflammatory syndrome, such as myocarditis.

The current Acute Coronary Syndrome guidelines lack specific recommendations for managing KS. Treatment involves prompt myocardial revascularization, although addressing the allergic reaction using second-generation antihistamines and corticosteroids should also be considered [14]. Vasodilators, including nitrates and calcium channel blockers, may alleviate vasospasm, but their use is limited in patients with shock due to the risk of exacerbating hypotension and tachycardia. Intracoronary administration of nitroglycerin might be a preferable option to mitigate vasospasm while avoiding systemic effects [2]. Caution is advised when using epinephrine in KS, as it can worsen cardiac ischemia and induce prolongation of QT interval and arrhythmias [14]. The right coronary artery is most commonly implicated, although the exact cause remains unclear [2]. Consistent with the literature data, all patients with Type II KS in our clinic presented with acute thrombosis of the right coronary artery or one of its branches.

Failure to identify KS may result in repeated allergen exposure, as it remains unrecognized as a potential trigger and, as such, possible recurrent acute coronary events. Diagnosis of KS should prompt complete allergy testing after the acute phase to guide treatment and preventive strategies [2]. Patients with known allergies or mast cell disorders are at increased risk of developing KS. Understanding the association enables clinicians to implement preemptive measures, such as alternative drugs, minimizing exposure or considering desensitization procedures in individuals with known hypersensitivity reactions. One case in the literature describes a patient with Type II KS who later tolerated exposure to the trigger after immunotherapy desensitization during the primary episode [15].

## 6. Future Directions and Research Gaps

Current ACS guidelines lack specific recommendations for managing KS, emphasizing the need for standardized protocols that combine allergological and cardiological assessments. Research should focus on safer diagnostic tools, such as in vitro assays, to identify triggers without provoking hypersensitivity reactions and explore the roles of non-IgE-mediated pathways and newer mediators, like platelet-activating factor. Validation of prophylactic interventions, such as mast cell stabilizers and desensitization protocols, is critical for high-risk patients. Additionally, studies are needed to evaluate KS involvement in non-coronary vascular territories, enhancing understanding of its systemic impact and guiding comprehensive risk assessment.

The true prevalence of Kounis syndrome is uncertain, with likely underreporting necessitating large-scale studies to better define its incidence and demographic patterns. Current treatment is based on ACS protocols, but comparative studies on tailored therapies, including antiplatelets, antihistamines, and corticosteroids, are needed to establish evidence-based guidelines. The long-term outcomes and predictors of recurrence remain poorly understood, highlighting the need for longitudinal research. Additionally, the role of advanced imaging and the integration of allergology and cardiology services require further exploration to optimize diagnosis, management, and patient outcomes.

## 7. Conclusions

This study enhances the understanding of Kounis syndrome by presenting four cases with varying triggers, clinical presentations, and outcomes, emphasizing its classification into three subtypes. The findings highlight the intricate relationship between allergic and cardiovascular mechanisms in KS, underlining the critical role of identifying allergic triggers as potential contributors to acute coronary events. Although rare, KS presents substantial diagnostic and therapeutic challenges due to its clinical overlap with conventional acute coronary syndromes.

The cases demonstrate the importance of timely diagnosis and comprehensive management, addressing both allergic and ischemic aspects, to mitigate significant morbidity. This research underscores the necessity for increased clinical vigilance, particularly in emergency care, and advocates for the integration of allergology and cardiology expertise in managing these patients. Further investigations are needed to develop standardized diagnostic frameworks, assess long-term outcomes, and evaluate prophylactic measures and customized treatment strategies for KS. These insights aim to advance clinical management of and stimulate continued research on this complex syndrome.

## Figures and Tables

**Figure 1 jcm-14-00768-f001:**
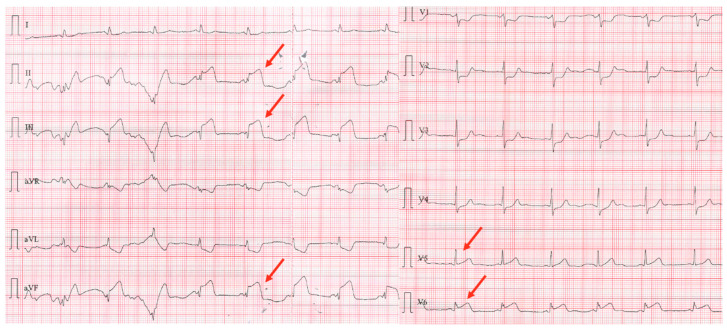
Electrocardiogram with postero-infero-lateral ST segment elevation myocardial infarction (the red arrows).

**Figure 2 jcm-14-00768-f002:**
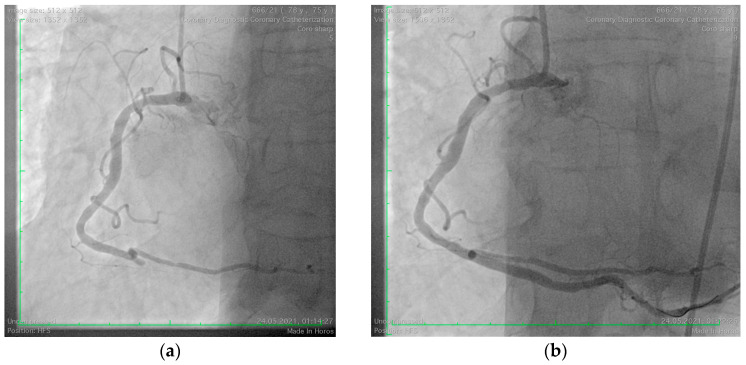
Coronary angiography of the right coronary artery. (**a**) Acute thrombotic closure of posterior interventricular artery; (**b**) Angiography aspect after revascularization by coronary angioplasty with a drug-eluting stent.

**Figure 3 jcm-14-00768-f003:**
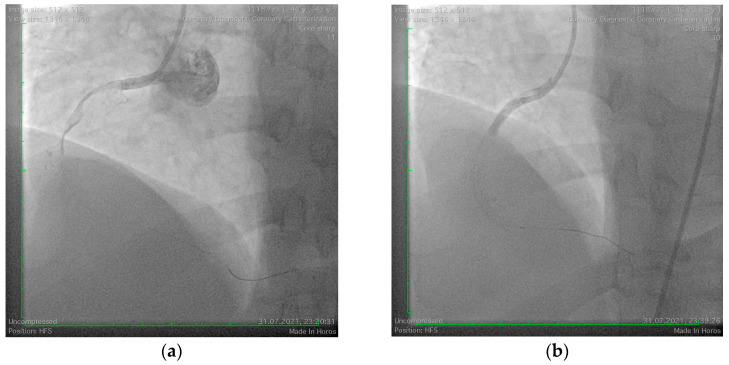
Coronary angiography of the right coronary artery. (**a**) Acute thrombotic occlusion of the proximal right coronary artery; (**b**) Angiography aspect after revascularization with phenomena of no reflow.

**Figure 4 jcm-14-00768-f004:**
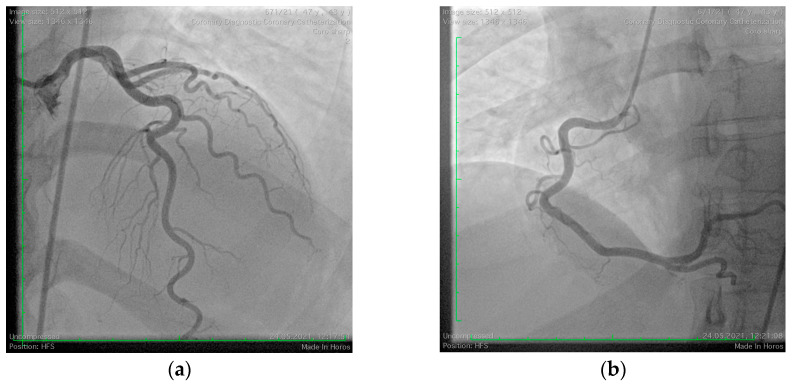
Coronary angiography with non-obstructive coronary arteries. (**a**) Normal left coronary arteries; (**b**) normal right coronary artery.

**Table 1 jcm-14-00768-t001:** Comparative characteristics of Type I, Type II, and Type III Kounis syndrome.

Characteristic	Type I Kounis Syndrome	Type II Kounis Syndrome	Type III Kounis Syndrome
**Pathophysiology and coronary anatomy**	Allergic vasospasm; normal coronary arteries	Allergen-induced rupture of atherosclerotic plaques	Hypersensitivity to stent materials causing inflammation and thrombosis
**Mechanism of injury**	Coronary vasospasm induced by mast cell degranulation, releasing histamine and leukotrienes	Plaque rupture due to enzymatic degradation by tryptase and chymase	Inflammatory response to stent components causing thrombus formation
**Immune mediators and triggers**	Histamine, leukotrienes, prostaglandins, tryptase, and platelet-activating factor; allergens (e.g., food, venom, toxins)	Histamine, leukotrienes, prostaglandins, tryptase, interleukins, tumor necrosis factor-alpha, and platelet-activating factor; immune activation exacerbating plaque vulnerability	Histamine, leukotrienes, tryptase, cytokines and platelet-activating factor; triggered by stent material hypersensitivity
**Clinical presentation and cardiovascular risk factors**	Transient chest pain due to vasospasm; reversible ischemia in younger, low-risk patients	Prolonged chest pain, ECG changes, and elevated cardiac biomarkers in patients with established cardiovascular risk factors and coronary artery disease	Sudden severe ischemia with in-stent thrombosis, observed in patients with known hypersensitivities to stent materials
**Diagnostic challenges and management**	Differentiated from non-ischemic causes of chest pain; treat with anti-allergic therapies, vasodilators (e.g., nitrates, calcium channel blockers)	Mimics acute myocardial infarction in patients with an allergen exposure history; manage with anti-inflammatory and anti-platelet therapies	Recognize hypersensitivity to stent materials presenting with ischemic symptoms; treat with antiplatelet and anti-inflammatory therapies and consider stent replacement if needed
**Distinctive aspects**	May involve non-IgE-mediated mast cell degranulation	Plaque instability driven by mast cell enzymes	Eosinophilic infiltration at the stent site amplifies thrombosis
**Prognosis and complications**	Usually reversible with allergen avoidance and treatment	High risk of myocardial infarction and ischemic complications	Significant risk of stent occlusion and major ischemic events

**Table 2 jcm-14-00768-t002:** Differential diagnosis between Takotsubo cardiomyopathy and Kounis syndrome.

Characteristic	Takotsubo Syndrome	Kounis Syndrome
**Epidemiology**	Postmenopausal women	Men aged between 40 and 70 years
**Pathophysiology**	Myocardial stunning caused by catecholamine surge triggered by emotional or physical stress	Allergen-mediated coronary events (vasospasm, plaque rupture or stent thrombosis)
**Clinical presentation**	Chest pain, dyspnea, symptoms mimicking acute coronary syndrome	Chest pain, dyspnea, often with allergic symptoms (angioedema, urticaria, anaphylaxis)
**Electrocardiogram and cardiac biomarkers**	ST-segment elevation, T-wave inversion, QT prolongation is common Troponin elevation disproportionate to dysfunction severity	ST-segment changes, T-wave inversion, less QT prolongation Troponin elevation indicating myocardial injury
**Echocardiography**	Extensive myocardial dysfunction beyond a single coronary territory; apical ballooning with akinesis/hypokinesis and hypercontractile basal segments Strain abnormalities are global, extending beyond the territory of a single coronary artery, often with significant apical involvement	Wall motion abnormalities localized to the affected coronary artery, resolving within days in post-acute phase Strain abnormalities are focal and correspond to coronary involvement
**Coronarography**	Normal coronary arteries (no obstructive lesions)	Vasospasm (Type I), plaque rupture (Type II), stent thrombosis (Type III)
**Prognosis**	Typically, full recovery within weeks	Depends on timely allergen avoidance and treatment

## Data Availability

All data generated or analyzed during this study are included in this article.

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
