# Peer review of "Kounis Syndrome in Clinical Practice: Insights from Clinical Case Series and Mechanistic Pathways"

_jcm, 2025, doi:10.3390/jcm14030768_

Round 1
Reviewer 1 Report
Comments and Suggestions for Authors
This article provides an overview of Kounis Syndrome (KS), explaining how allergic reactions can trigger cardiac events. The authors clearly present the three types of KS and their mechanisms, emphasizing the role of mast cells and inflammation. Including case studies illustrates the diverse presentations and management of KS. The article also discusses management strategies, including a multidisciplinary approach and various medications. The article highlights the need for future research, such as safer diagnostic tools and preventive interventions. Overall, this article is a valuable resource for clinicians seeking to enhance their understanding of Kounis Syndrome.
Author Response
Dear Reviewer,
Our team would like to sincerely thank you for the positive manuscript appreciation.
Best regards,
All authors

Reviewer 2 Report
Comments and Suggestions for Authors
In this interesting case series, the authors discussed Kounis Syndrome. Overall, this is an insightful manuscript. However, some amendments must be done to improve the readability of this manuscript:
1) The presentation has to be improved. Avoid using bullet points in an article (unless this is a lab report). Instead, convert them into nicely flowing paragraphs.
2) "Kounis Syndrome (KS) is a complex clinical entity representing the intersection of allergic responses and acute coronary syndromes (ACS)." What does this mean? So is it ACS? or it is not an outcome of an allergic reaction?
3) There are many short paragraphs that can be combined into longer ones. Consider revising.
4) Line 53: I suggest improving the study aims.
5) "Upon allergen exposure, mast cells undergo degranulation, releasing mediators such as histamine, leukotrienes, and platelet-activating factor (PAF). " The authors may want to include some info from this publication (https://www.sciencedirect.com/science/article/pii/S2949907024000044)
6) "Furthermore, emerging evidence highlights the involvement of non-IgE pathways in mast cell activation, demonstrating that allergens can directly stimulate mast cell degranulation even in the absence of IgE antibodies." So, I assume this process is mainly through non-IgE mediated type 1 hypersensitivity reaction aka pseudoallergy? Please explain.
7) Add some information about the prevalence of this disease globally. How often does it occur during allergic reaction?
8) "Kounis Syndrome encompasses a complex interplay of immunological mechanisms, involving both IgE-mediated and non-IgE-mediated pathways." When is it IgE-mediated and when it is not? Please explain more clearly
9) Sections 2.1 and 2.2 seem to contain redundant information. Consider merging them.
10) The authors can add a table comparing KS I, II and III to emphasize their differences even more.
11) How often is this KS accompanied with other organ manifestations of allergy, for instance skin rash or angioedema?
12) Please explain the differences between KS III and common stent thrombosis? Is all stent thrombosis due to KS type III?
13) In Case 1, how did the authors confirm that indeed the bee sting caused KS type II? Did the authors check the serum tryptase? IgE? Eosinophil? I don't see any allergy medication given so why bother diagnosing with KS?
14) Similarly, the argument of saying that case 2 was due to KS is not strong enough. The thrombus was confirmed through coronary angiography but the connection with the antibiotics is still arguable.
15) Similarly, case 3 has the same issue.
16) I do think that maybe it is better to rewrite this article as a narrative review rather than case series as the cases are not confirmed yet. Otherwise, the authors need to make sure that those "allergens" are definite causes of the CAD/ACS in those cases.
17) The presentation must also be improved. It is confusing to read the case in the middle of the review. It is better to present the introduction (including classification), then cases, then review. But as I said above, those supporting data must be included if the authors want to keep the cases.
Author Response
Dear Reviewer,
Our team would like to sincerely thank you for the positive manuscript appreciation and for all your valuable suggestions. We have carefully addressed your comments and have revised our manuscript accordingly. Revised portions are highlighted in the text.
We hope these responses clarifies your concerns, and we remain available for any further questions.
Best regards,
All authors

Reviewer 3 Report
Comments and Suggestions for Authors
You have conducted research on the pathophysiology and treatment of the disease by presenting four cases of KS, a rare disease. The abstract section is well organized with a framework of purpose, method, results, and conclusion, but the main text does not have a general paper format. Of course, there is a specificity in presenting rare cases, but please review whether this paper can be reorganized within the framework of introduction, method, results, discussion, and conclusion. In addition, please revise and supplement the content below.
. Please describe the previous research in the introduction section.
. There is no research method in the framework of the paper, so please describe the design and research subjects of this study in one place.
. Lastly, the conclusion section is too short. Please describe the research contents concisely and mention the implications and applications of the research results.
Author Response

(The authors gave the same response as above.)

Round 2
Reviewer 2 Report
Comments and Suggestions for Authors
Thanks for the response. I do have further comments:
1. A review article does not need a structured abstract. Please amend
2. Please provide the non-tracked version of the article. I think by deleting the section head of 2.1, now the introduction is lengthy.
3. In Table 1, row "immune mediators", tryptase was not mentioned in KS I and KS III, why? Please check whether those mediators are accurately specified.
4. The order of listing in Table 1 is confusing. What is the difference between histological features and histopathological evidence? Also, prognostic implications vs. Prognosis and complications? This table seems to be AI generated. Is this true?
5. In a review article, methods, results and discussion is not relevant. Please revise.
6. In Page 10, the authors compared Takotsubo with Kounis, but do they actually look alike and clinically indistinguishable? I doubt it. Is there apical balloning in KS? What about the strain pattern? the GLS? what about the findings from speckle tracking echo? Is it true that the pathognomonic symptom of TTS is chest pain as seen in KS? I do think that KS type I is resembling prinzmetal angina though.
7. Try to compare those two (TTS vs KS) in a table.
I think in general, the authors need to take their time to improve the accuracy of the content and the presentation of this manuscript.
Author Response
Dear Reviewer,
Our team would like to sincerely thank you for all your valuable suggestions. We have carefully addressed your comments and have revised our manuscript accordingly. Revised portions are highlighted in the text.
Please see the attachment.
Kind regards,
All authors

Reviewer 3 Report
Comments and Suggestions for Authors
Thank you for your efforts.
Author Response

(The authors gave the same response as above.)
